# Lactobacillus Plantarum 108 Inhibits Streptococcus mutans and Candida albicans Mixed-Species Biofilm Formation

**DOI:** 10.3390/antibiotics9080478

**Published:** 2020-08-04

**Authors:** Neha Srivastava, Kassapa Ellepola, Nityasri Venkiteswaran, Louis Yi Ann Chai, Tomoko Ohshima, Chaminda Jayampath Seneviratne

**Affiliations:** 1Oral Sciences, Faculty of Dentistry, National University of Singapore, Singapore 11908, Singapore; E0056315@u.nus.edu (N.S.); or kellep@lsuhsc.edu (K.E.); E0011745@u.nus.edu (N.V.); 2Center of Oral and Craniofacial Biology, School of Dentistry, Louisiana State University Health Sciences Center, New Orleans, LA 70112, USA; 3Division of Infectious Diseases, University Medicine Cluster, National University Health System and Faculty of Medicine, Yong Loo Lin School of Medicine, National University of Singapore, Singapore 117597, Singapore; louis_chai@nuhs.edu.sg; 4Department of Oral Microbiology, School of Dental Medicine, Tsurumi University, Yokohama 230-8501, Japan; ohshima-t@fs.tsurumi-u.ac.jp; 5Singapore Oral Microbiomics Initiative, National Dental Research Institute Singapore (NDRIS), National Dental Centre Singapore, SingHealth Duke NUS Medical School, 5 Second Hospital Avenue, Singapore 168938, Singapore

**Keywords:** *Streptococcus mutans*, Lactobacillus plantarum, glucosyltransferase, dental caries, probiotics

## Abstract

*Streptococcus mutans* is the principal biofilm forming oral pathogen associated with dental caries. Studies have shown that *Candida albicans*, a commensal oral fungus is capable of forming pathogenic mixed-species biofilms with *S. mutans.* The treatment of bacterial and fungal infections using conventional antimicrobial agents has become challenging due to the antimicrobial resistance of the biofilm mode of growth. The present study aimed to evaluate the efficacy of secretory components of *Lactobacillus plantarum 108*, a potentially promising probiotic strain, against *S. mutans* and *C. albicans* single and mixed-species biofilms. *L. plantarum 108* supernatant inhibited *S. mutans* and *C. albicans* single-species biofilms as shown by XTT reduction assay, crystal violet assay, and colony forming units counting. The probiotic supernatant significantly inhibited the *S. mutans and C. albicans* mixed-species biofilm formation. The pre-formed mixed-species biofilms were also successfully reduced. Confocal microscopy showed poorly developed biofilm architecture in the probiotic supernatant treated biofilms. Moreover, the expression of *S. mutans* genes associated with glucosyltransferase activity and *C. albicans* hyphal specific genes (*HWP1, ALS1* and *ALS3*) were down-regulated in the presence of the probiotic supernatant. Altogether, the data demonstrated the capacity of *L. plantarum 108* supernatant to inhibit the *S. mutans* and *C. albicans* mixed-species biofilms. Herein, we provide a new insight on the potential of probiotic-based strategies to prevent bacterial-fungal mixed-species biofilms associated with dental caries.

## 1. Introduction

Early childhood caries (ECC), a major oral health problem worldwide is an aggressive form of dental caries that affects children under six years of age. According to World Health Organization, the prevalence of ECC is reported to be 17% among two-year old children, 48% among four-year old children and 70% among six-year old children [1]. ECC is related to prolonged intake of dietary sugars that basically comprises of sucrose [2]. Intake of sugary beverage through nipple bottles is very common in children. This increases the adverse effect of sucrose and limits buffering saliva to reach tooth surface [3,4]. Pathogenic bacteria form a biofilm on dental tissues and produce sugar driven acids. This dissolves the mineral structure of the teeth irreversibly, thereby causing dental caries [5]. If left untreated, ECC may lead to cavitation and subsequent pulpal infection causing severe pain which requires expensive treatment [6].

*Streptococcus mutans*, an acidogenic bacterium forming a virulent plaque biofilm on the tooth surface is considered a key pathogen associated with ECC [7,8]. *S. mutans* is able to rapidly utilize fermentable dietary sucrose and synthesize extracellular glucans through several exoenzymes, such as glucosyltransferases (Gtfs). This extracellular glucan enhances the bacterial adhesion to the tooth surface, as well as aid in bacterial co-aggregation leading to the development of highly virulent mixed-species biofilms in the oral cavity [9]. These complex biofilms facilitate the development of an acidogenic microenvironment, subsequently demineralising the tooth enamel [10].

Previous studies have demonstrated that *S. mutans* forms mixed-species biofilms with the human fungal pathogen *Candida albicans* in dental plaque [11,12]. *S. mutans* derived Gtfs bind firmly to the cell surface of *C. albicans* [13]. Adhesion between the foregoing microorganisms has been shown to be significantly enhanced in the presence of sucrose [14,15,16]. Gtfs bound to the *C. albicans* cell surface produce large amounts of glucans in the presence of sucrose. These glucans in turn increase the binding sites for *S. mutans* [13,17] resulting in a highly pathogenic mixed-species biofilm [15]. We demonstrated that *S. mutans* derived GtfB is able to augment the accumulation of *C. albicans* in mixed-species biofilms and enhance *C. albicans* growth by cross-feeding of glucose and fructose by sucrose breakdown [18,19]. Moreover, *S. mutans* was able to up-regulate the expression of *C. albicans* hypha associated genes such as *HWP1*, *ALS1* and *ALS3* which can be attributed to an increase in virulence of the organism in mixed-species biofilms. On the other hand, *C. albicans* also enhanced the development of *S. mutans* microcolonies in mixed-species biofilms [20]. Taken together, *S. mutans* and *C. albicans* demonstrate a symbiotic relationship in the mixed-species biofilm with complex inter-species interactions. Thus, a complex microenvironment such as the dental plaque displays resistance to most antimicrobials [21,22].

Use of probiotics as a strategy for controlling dental plaque biofilms has recently gained significant interest [23,24]. According to the 2001 definition by the World Health Organization, probiotics are “live microorganisms which, when administered in adequate amounts, confer a health benefit on the host” [25]. The oral cavity is a rich and diverse ecosystem inhabited by both bacteria and fungi, collectively occupying and co-existing within various niches as biofilm communities [26]. In the ecological perspective, dental caries is a result of dysbiosis of the oral flora and probiotics can be used for controlling microbial changes in the oral environment. However, when administration of live microorganisms may not be an ideal therapeutic option under certain conditions, application of their secretory components that demonstrate the ability to support the growth of healthy bacterial species and inhibit pathogenic species can be a promising strategy. As a result, focus has shifted towards testing the antimicrobial activity of secretory components from probiotic bacteria [27]. Probiotic bacteria are known to produce various antimicrobial compounds that can efficiently inhibit bacterial adhesion and disrupt biofilm formation [28]. Studies have demonstrated the promising anti-biofilm activity of probiotic-based strategies against *S. mutans* and *C. albicans* single-species biofilms [24,29,30,31]. The probiotic activity of *Lactobacillus* species such as *Lactobacillus salivarius* and *Lactobacillus rhamnosus* strains were evaluated against a *S. mutans-C. albicans* dual-species biofilms and a multi-species biofilm model, respectively [32,33]. The present study aimed to evaluate the efficacy of the secretory supernatant of a promising probiotic strain, *Lactobacillus plantarum 108* (Lp108) against *S. mutans* and *C. albicans* mixed-species biofilms.

## 2. Results

### 2.1. Effect of Lactobacillus plantarum 108 Supernatant on Planktonic Streptococcus mutans and Candida albicans

Growth kinetic assay showed that Lp108 supernatant has a significant inhibition on the planktonic mode of growth of both *S. mutans* and *C. albicans* compared to the control group without supernatant. In control groups, *S. mutans* and *C. albicans* cells showed an exponential growth around five hours after incubation, whereas much lesser growth was detected in the group treated with supernatant (Figure 1).

### 2.2. Effect of Lactobacillus Plantarum 108 Supernatant On Streptococcus Mutans Biofilm Formation

*S. mutans* was cultured in the presence and absence of the probiotic supernatant and biofilm formation was quantified using Tetrazolium salt 2,3 - bis (2 - methoxy - 4 – nitro - 5- sulfophenyl) -5- [(phenylamino) carbonyl]-2*H*- tetrazolium hydroxide (XTT) reduction assay [34], Crystal Violet (CV) assay [35], and Colony forming Unit (CFU) counting. When the supernatant was supplemented at the beginning of incubation (0 h), XTT assay showed a significant reduction (89%) in the activity of *S. mutans* in the presence of Lp108 supernatant compared to the control (Figure 2A; *p* < 0.05). Similarly, CV assay demonstrated that total biofilm biomass was reduced by 73% when treated with probiotic supernatant (Figure 2B; *p* <0.05). CFU counting demonstrated 99% reduction in cell numbers in the test group compared to the untreated control (Figure 2C; *p* < 0.05).

Activity of probiotic supernatant on pre-formed *S. mutans* biofilms was also evaluated. An inhibition of *S. mutans* biofilms by 33%, 36% and 94% was observed with XTT assay, CV assay and CFU assay respectively, following treatment with probiotic supernatant after 12 h of initial incubation (Figure 2; *p* < 0.05). These results suggest that the Lp108 supernatant was not only able to inhibit the initial colonization of *S. mutans* biofilm, but also eradicates the preformed biofilms.

### 2.3. Effect of Lactobacillus Plantarum 108 Supernatant on Candida Albicans Biofilm Formation

There was a significant inhibition of *C. albicans* biofilm formation when treated with Lp108 supernatant i.e., 90%, 80% and 91% inhibition in XTT assay, CV assay, and CFU counting, respectively (Figure 3; *p* < 0.05). Probiotic supernatant also effectively eradicated the preformed *C. albicans* biofilms. The XTT assay, CV assay and CFU counting demonstrated a reduction by 34%, 42% and 46 % respectively (Figure 3; *p* < 0.05).

### 2.4. Effect of Lactobacillus plantarum 108 Supernatant on Streptococcus mutans and Candida albicans Mixed-Species Biofilms

Interestingly, a significant inhibition was observed in mixed-species biofilm formation when the Lp108 supernatant was supplemented at the beginning of the incubation. The XTT reduction assay showed 85% reduction in the mixed-species biofilm (Figure 4A; *p* < 0.05) whereas CV assay also showed 86% reduction (Figure 4B; *p* < 0.05) in total biofilm biomass of the treated mixed-species biofilms. The CFU counting method showed a significant reduction in both *S. mutans* and *C. albicans* cells in the probiotic-treated biofilm samples compared to the control without supernatant. The probiotic supernatant was able to reduce the *S. mutans* and *C. albicans* cells by 99.99% and 99.34% respectively in the mixed-species biofilm (Figure 4C,D; *p* < 0.05).

Further, Lp108 supernatant was also observed to eradicate the preformed *S. mutans* and *C. albicans* mixed-species biofilms. The XTT assay, and CV assay showed a reduction in the preformed mixed-species biofilms by 33% and 50% respectively (Figure 4A,B; *p* < 0.05). In the presence of probiotic supernatant, cultivable *S. mutans* and *C. albicans* cells were reduced by 91.50% (Figure 4C; *p* < 0.05) and 43.68% (Figure 4D) respectively. Foregoing data demonstrate that Lp108 supernatant was effective against *S. mutans* and *C. albicans* mixed-species biofilms.

### 2.5. Structural Analysis of Biofilms by Confocal Laser Scanning Microscopy

Confocal Laser Scanning Microscopy (CLSM) was used to further validate the efficacy of Lp108 supernatant against *S. mutans* and *C. albicans* single and mixed-species biofilms. CLSM images of the control groups showed a dense accumulation of *S. mutans* (Figure 5A) and *C. albicans* (Figure 5D) cells in the single-species biofilm showing the typical biofilm architectures. Closely aggregated *S. mutans* and *C. albicans* yeast cells clustering together with intermittent hyphal distribution was observed in the *S. mutans*-*C. albicans* mixed-species biofilm control (Figure 5G). On the contrary, supplementation of probiotic supernatant significantly inhibited the biofilm formation of *S. mutans* and *C. albicans* single and mixed-species biofilms (Figure 5B,E,H). However, preformed biofilms treated with the Lp108 supernatant comparatively had more cells than the biofilms of the inhibitory experiment assay (Figure 5C,F,I). CLSM observations were consistent with the data obtained from the biofilm quantification assays.

### 2.6. Lactobacillus plantarum 108 Supernatant Down-Regulated the Gtf Gene Expression in Streptococcus mutans Single and Mixed-Species Biofilms

qRT-PCR analysis of the single-species biofilm demonstrated that the Lp108 supernatant significantly down-regulated the expression of genes associated with Gtf activity of *S. mutans*. Expression of *gtfB, gtfC,* and *gtfD* was downregulated by 48.8%, 44.7%, and 65.7% respectively as compared to the untreated control groups (Figure 6A; *p* < 0.05). Mixed-species biofilms also showed downregulation in the expression of *gtfB*, *gtfC,* and *gtfD* by 21.8%, 29.3%, and 35.6%, respectively as compared to their corresponding untreated control group (Figure 6B; *p* < 0.05). Relative fold change compared to control and *p* values are summarized in Appendix A.

### 2.7. Lactobacillus plantarum 108 Supernatant Down-Regulated the Expression of HWP1, ALS1 and ALS3 Genes in Candida albicans Single and Mixed-Species Biofilms

The expression of *HWP1, ALS1* and *ALS3* genes in the single-species *C. albicans* biofilm treated with Lp108 supernatant was significantly down-regulated with respect to the untreated control groups. *HWP1, ALS1* and *ALS3* genes were down-regulated by 84.3%, 84.4% and 72.9% respectively (Figure 7A; *p* < 0.05).

Down-regulation of hyphal growth associated genes *HWP1* (58.3%)*, ALS1* (33.9%) and *ALS3* (39.1%) were also observed in the mixed-species biofilm group treated with probiotic supernatant (Figure 7B; *p* < 0.05). Relative fold change compared to control and *p*-values are summarized in Appendix A. Gene expression results were found to be consistent with the inhibitory activity of probiotic supernatant shown by biofilm quantification assays and confocal imaging.

## 3. Discussion

Probiotic based strategies have shown promising results in inhibiting *S. mutans* and *C. albicans* biofilms. Herein we observed that cell-free supernatant from a new probiotic *L. plantarum* strain (Lp108) inhibits not only the initial colonization but also the preformed *S. mutans* and *C. albicans* single as well as mixed-species biofilms. Lp108 supernatant was most effective in inhibiting the early phase of biofilm formation and had comparatively reduced activity against the pre-formed biofilms. Numerous *in-vitro* studies have shown that different probiotic strains inhibit *S. mutans* [24,31,36,37] and *C. albicans* [29,38,39,40] single-species biofilm formation. Clinical studies have also shown the inhibitory capacity of various probiotics against *S. mutans* [41,42] and *C. albicans* [43,44,45] in patients suffering from oral health problems.

The antimicrobial effect of the probiotic supernatant could be attributed to the presence of a mixture of antimicrobial peptides and other antimicrobial compounds in the secretome of *L. plantarum 108*. Previous studies have attempted to purify secretory components from other *L. plantarum* strains. A secreted low molecular weight compound designated as plantaricin was isolated from a *L. plantarum* LR/14 strain and was found to have strong bactericidal characteristics, heat stability and tolerance to acids [46]. Subsequently, plantaricin was found to be proteinaceous in nature and exerted its antifungal activity through leakage of intra-cellular contents leading to cell death [47]. Hence, it is likely that the *L. plantarum* supernatant used in the present study may also possess molecules with similar properties; however other mechanisms of action cannot be ruled out. For example, the nature of the interaction of *L. plantarum* supernatant with biofilms might be physiochemical. It can be assumed that secretory component of *L. plantarum* supernatant might have modified the surface energies of the microorganisms and inhibited complex biofilm formation by preventing the microbial coaggregation [48].

Furthermore, as shown by CLSM images, Lp108 supernatant may also have molecules that inhibit the adhesion of *S. mutans* and *C. albicans* to solid surfaces. This can be attributed to the biosurfactants and exometabolites in the supernatant that account for reduction in the hydrophobicity of surface substratum by interfering with microbial adhesion and desorption processes [49]. Similar findings have been reported in other studies wherein biosurfactants reduce the microbial adhesion to solid surfaces [48,50,51,52].

Production of glucans from *S. mutans* is regarded as a crucial virulence factor in the pathogenesis of dental caries [9,53]. Interestingly, our findings revealed that Lp108 supernatant significantly down-regulated the expression of all three *gtf* genes i.e., *gtfB, gtfC, and gtfD* in *S. mutans* single and mixed-species biofilms. Regulatory mechanisms of genes encoding Gtf enzymes in *S. mutans* is complex and has not been fully elucidated. However, it can be assumed that the active components in the probiotic interfere with the Gtf enzymes production at the gene expression level. As a result, it reduces the *S. mutans* attachment and biofilm formation. There are also other studies that demonstrate the ability of probiotic strains to down regulate *gtf* genes in *S. mutans* [31,54,55,56].

*ALS3* and *ALS1* genes belong to the *ALS* family of adhesins which are generally over expressed during in vitro adhesion of *C. albicans* to the epithelial cells [57,58]. The *HWP1* is known to encode the *C. albicans* protein which is responsible for the maintenance of cell wall integrity, hyphal development and intracellular signaling [57]. *HWP1* and *ALS3* mutants of *C. albicans* are defective in biofilm formation [59,60]. Recently, we demonstrated that *S. mutans* derived GtfB is able to up-regulate the expression of these genes of *C. albicans* in mixed-species biofilms [18]. Interestingly, in this study, we found that probiotic supernatant significantly down regulated the expression of *HWP1, ALS1* and *ALS3* of *C. albicans* in single and mixed-species biofilms. Our results corroborated previous studies that reported down regulation of these genes after treatment of *C. albicans* with probiotic strains [29,61]. Foregoing findings explain the inhibitory effect of Lp108 supernatant on *S. mutans* and *C. albicans* single and mixed-species biofilms.

In conclusion, the results of the present study indicate that Lp108 supernatant was able to inhibit the *S. mutans* and *C. albicans* single and mixed-species biofilms. Furthermore, structural analysis through confocal imaging provided evidence for the ability of secretory components of *L. plantarum 108* to inhibit the development of biofilm architecture. Finally, through gene expression analysis we were able to demonstrate a significant down-regulation of the genes responsible for glucosyltransferase activity of *S. mutans* and also that of genes involved in yeast-hypha transition in *C. albicans*. However, further studies are required to decipher the exact antimicrobial compound/s of this probiotic supernatant that inhibits *S. mutans* and *C. albicans*. If proven feasible, probiotic-based anti-biofilm strategies will be highly useful to treat bacterial-fungal mixed-species biofilm infections, including ECC.

## 4. Materials and Methods

### 4.1. Microbial Strains and Culture Conditions

*L. plantarum 108* strain was identified in a study which screened potential probiotic isolates from human oral cavities at the Tsurumi University School of Dental Medicine [37]. *S. mutans UA159*, originally isolated from a child with active caries and the fungal strain *C. albicans SC5314*, a clinic strain originally isolated from a patient with generalized *Candida* infection were revived from the archival collection stored at -80 °C in the School of Dentistry, National University of Singapore. Standard cell suspensions were prepared in ultra-filtered tryptone yeast extract medium (UFTYE) with 1% (*w*/*v*) glucose at pH 7 and pH 5.5 for *S. mutans* and *C. albicans* respectively, according to a previously established protocol [18]. Overnight cultures were centrifuged (4000× *g*, 10 min, 4 °C) and washed twice in phosphate-buffered saline (PBS). Subsequently optical densities (OD) of the *S. mutans* (OD 0.300) and *C. albicans* (OD 0.375) cell suspensions were adjusted at a wave length of 520 nm, using a spectrophotometer (UV-1700 Shimadzu, Kyoto, Japan).

### 4.2. Preparation of Lactobacillus plantarum 108 Supernatant

*L. plantarum 108* cells were grown for 18 h in de Man, Rogosa and Sharpe (MRS) broth, prepared by adding 27.5 g of MRS (Sigma-Aldrich) to 500 mL of distilled water and the broth was sterilized by autoclaving. This overnight culture was centrifuged (4000× *g*, 4 °C, 10 min), washed twice and resuspended in PBS and the optical density of the standard cell suspension was adjusted (OD 0.5 at 600 nm) using a spectrophotometer (UV-1700 Shimadzu, Kyoto, Japan). For supernatant preparation, this standard cell suspension was diluted 1:100 in MRS broth and further incubated for 24 h at 37 °C. The bacterial culture was subjected to centrifugation (4000× *g*, 4 °C, 10 min), and subsequently the supernatant was filtered through a 0.22 µm pore size membrane (Surfactant-free cellulose acetate, Ministart syringe filter, Sartorius, Singapore). This standardized supernatant was used in all the experiments [37].

### 4.3. Antimicrobial Activity against Planktonic Streptococcous mutans and Candida albicans

Planktonic cells of *S. mutans* (2 × 10^6^ CFU/mL) and *C. albicans* (1 × 10^6^ CFU/mL) were incubated with UFTYE broth containing 2% (*w*/*v*) sucrose [15], in 96-well microtiter plates in the presence and absence of Lp108 supernatant. Subsequently, the plate was incubated for 24 h at 37 °C and the optical density was measured at a wavelength of 600 nm using a spectrophotometer (UV-1700 Shimadzu, Kyoto, Japan). The growth was measured at 30 min intervals for 24 h.

### 4.4. In Vitro Biofilm Formation and Treatment of Biofilms with Lactobacillus plantarum 108 Supernatant

For *S. mutans* and *C. albicans* single-species biofilm formation, a cell suspension containing a cell concentration of 1 × 10^6^ CFU/mL yeast cells or 2 × 10^6^ CFU/mL bacterial cells were inoculated in UFTYE broth containing 2% (*w*/*v*) sucrose. For mixed-species biofilm formation, equal volumes of 1 × 10^6^ CFU/mL yeast cells and 2 × 10^6^ CFU/mL bacterial cells with UFTYE broth containing 2% (*w*/*v*) sucrose were pipetted into each well of a microtiter plate according to a previously established protocol [18].

Two assays were conducted to examine the anti-biofilm activity of cell-free *L. plantarum 108* supernatant. Firstly, Lp108 supernatant was introduced into the wells at the beginning (0 h) along with bacterial or fungal cell suspensions to evaluate the preventive ability of the supernatant against biofilms. For the therapeutic assay, respective *S. mutans* and *C. albicans* biofilms were formed for 12 h and subsequently the probiotic supernatant was added to these preformed biofilms. Similarly, for mixed-species biofilms probiotic supernatant was added at two different time points (0 h and 12 h).

### 4.5. Quantification of Biofilms

Biofilms were quantified using three different techniques, namely XTT reduction assay, crystal violet (CV) assay and colony forming units (CFU) counting. XTT assay quantifies the metabolic activity of the biofilms, CV assay estimates the total biomass of biofilms including extracellular matrix, whereas CFU assay indicates the number of cells in the biofilm.

#### 4.5.1. XTT Reduction Assay

Biofilm formation was evaluated using tetrazolium salt XTT reduction assay as previously described [21]. In brief, after removing the culture media, biofilms were washed once with PBS to remove non-adherent cells. Subsequently, 200 μL of XTT solution (4 μM menadione and 0.2 mg/mL XTT in PBS) was added to each well and incubated in the dark for 20 min at 37 °C. Colorimetric changes were measured by using a plate reader at 490 nm (Multiskan™ GO, Thermo Scientific, Singapore).

#### 4.5.2. Crystal Violet Assay

Crystal Violet (CV) assay was performed according to a previously published protocol with minor modifications [35]. In brief, biofilms were washed with PBS once, air dried and fixed in 2% formalin. Subsequently, biofilms were stained with 1% (*w*/*v*) of CV for 5 min. After washing, the plates were air dried and an aliquot of 200 μL of 95% ethanol was added and incubated for 15 min. Optical density of 95% ethanol was measured at 570 nm (Plate reader, Multiskan™ GO, Thermo Scientific, Singapore).

#### 4.5.3. Colony-Forming Units Counting

A 10-fold dilution series of the cell suspensions from both single and mixed-species biofilms was prepared in PBS. Different dilutions of *C. albicans* and *S. mutans* were spread on glucose minimal medium (GMM) agar (6.79 g/l yeast nitrogen base without amino acids; Difco, 2% glucose and 1% agar; Sigma-Aldrich) and brain-heart infusion (BHI; Difco) agar plates, respectively. For mixed species, GMM and BHI plates were supplemented with 8 μg/mL gentamicin sulfate salt to prevent bacterial growth and 8 μg/mL of amphotericin B to prevent fungal growth. The BHI plates were incubated at 37 °C for 24 h and the GMM plates were incubated at 30 °C for 48 h. The bacterial and fungal colonies were counted and the corresponding log CFU values were calculated.

### 4.6. Confocal Laser Scanning Microscopy

Single and mixed-species biofilms were formed on 8-well chamber slides (Nunc, Thermo Scientific, Lab-Tek™, Singapore) and the test samples were treated with Lp108 supernatant. For confocal laser scanning microscopy (CLSM), biofilms were stained according to a previously described protocol [18]. In brief, biofilms were fixed with 4% (*v*/*v*) paraformaldehyde and stained with 200 μL of propidium iodide (Invitrogen, Thermo Fisher Scientific, Singapore) for *S. mutans* and 0.001% (*w*/*v*) calcofluor white (Sigma-Aldrich, Singapore) for *C. albicans*. The plates were incubated for 20 min in dark and thereafter washed with PBS. Biofilms were visualized using an Olympus-Fluoview FV1000 TIRF confocal microscope. Z-sections were obtained from representative microscopic fields of three biological replicates. The single and mixed-species biofilm architecture was analysed using Imaris software.

### 4.7. Gene Expression Analysis by qRT-PCR

The expression of genes associated with *S. mutans* Gtf enzyme activity such as *gtfB, gtfC* and *gtfD* [9] as well as the hyphal growth associated genes in *C. albicans* such as *HWP1, ALS1* and *ALS3* were evaluated using quantitative real-time PCR (qRT-PCR) [62]. Biofilm samples were prepared as described above. After incubation, cells from control and probiotic supernatant treated biofilms were harvested and the cell pellets were collected by centrifugation (10,000× *g* for 10 min). TRIzol reagent (Invitrogen Ambion, Singapore) was used to extract the total RNA from cell pellets according to a previously published protocol [63]. NanoDrop ND 1000 spectrophotometer (Thermo Scientific, Singapore) was used to determine the concentration, purity and quality of the isolated RNA by measuring the absorbance ratio at 260/280 nm and 260/230 nm. RNA was reverse transcribed into cDNA using the M-MLV Reverse Transcriptase system (Promega, Singapore). A list of primers used for amplifying the target and housekeeping genes used are given in the Table 1 and Table 2. Diluted cDNA, gene specific forward and reverse primers and SYBR Green (KAPA SYBR FAST qPCR Kit, Kapa Biosystems, Wilmington, MA, USA) were mixed in reaction mixture and qRT-PCR was performed using the Step One Plus^TM^ Real-Time PCR system (Thermo Fisher Scientific, Singapore). This was done under adequate thermo cycling conditions (Holding stage 95 °C for 3min, cycling stage 95 °C for 3 s, 60 °C for 1 min and Melting curve stage 95 °C for 1 s, 60 °C for 1 min and 95 °C for 15 s) for 40 cycles. The resultant C_T_ values of the target genes of interest were normalized to the C_T_ values of the respective housekeeping gene (*16sRNA* for *S. mutans* and *PMA1* for *C. albicans*). Results were analyzed using the $2^{-\Delta \Delta C_{t}}$ relative expression method to calculate the fold changes [64].

### 4.8. Statistical Analysis

All experiments were carried out in triplicate and on three different occasions. The obtained data was expressed as mean values with the corresponding standard deviations (SD). For pair wise comparison, Student’s *t* test or Mann-Whitney *U* test was performed on all data sets to compare treated groups with respect to control groups and *p* < 0.05 was considered statistically significant (*). SPSS software version 20.0 was used for the statistical analysis.

## 5. Conclusions

The present study demonstrated the ability of *L. plantarum 108* supernatant to inhibit *S. mutans* and *C. albicans* single and mixed-species biofilm formation

## Figures and Tables

**Figure 1 antibiotics-09-00478-f001:**
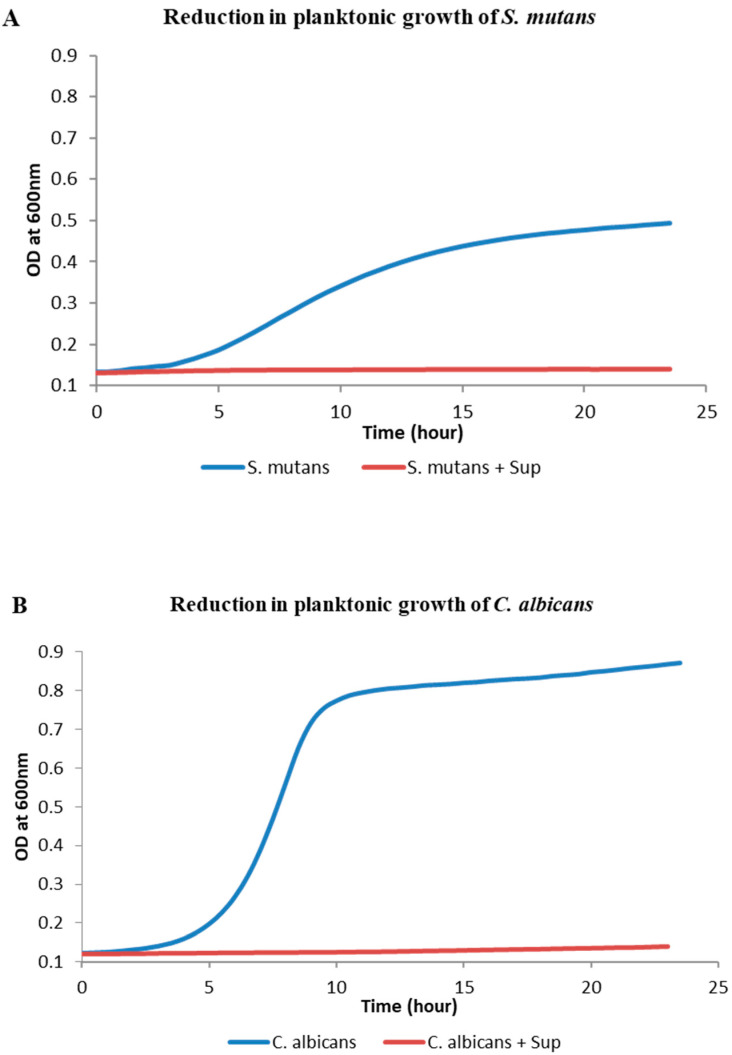
*Lactobacillus plantarum 108* supernatant (Sup) inhibited the planktonic growth of *S*. *mutans* and *C. albicans. S. mutans* (**A**) and *C. albicans* (**B**) growth evaluated using optical density measurements demonstrated a significant inhibition of growth when supplemented with the supernatant.

**Figure 2 antibiotics-09-00478-f002:**
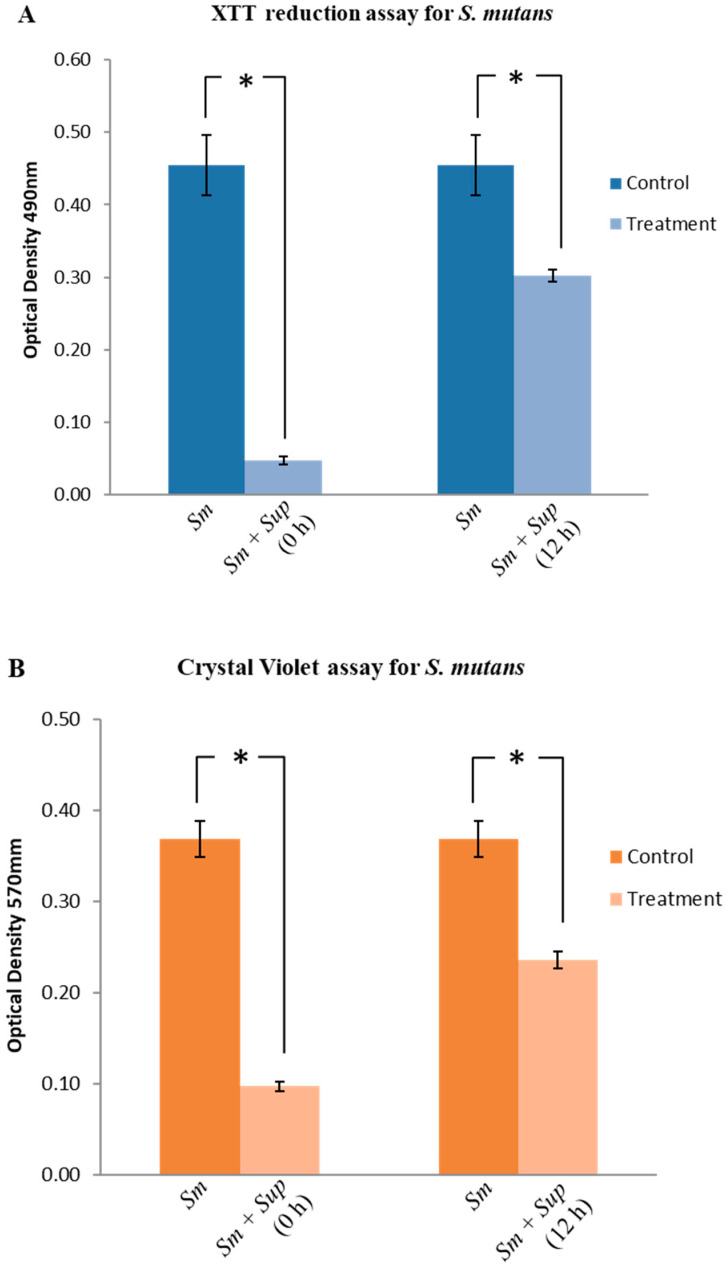
*Lactobacillus plantarum 108* supernatant (Sup) inhibited *S. mutans* (Sm) biofilm formation and demonstrated activity against preformed biofilms. Lp108 supernatant was introduced at two time points; 0 h for the preventive assay and at 12 h into a preformed biofilm for the therapeutic assay. Biofilms were quantified using (**A**) XTT reduction assay, (**B**) Crystal Violet assay and (**C**) Colony forming unit counting (CFU). Data are presented as mean ± SD and statistical significance indicated by the asterisk (*) was evaluated with respect to the control group without the Lp108 supernatant treatment (*p* < 0.05).

**Figure 3 antibiotics-09-00478-f003:**
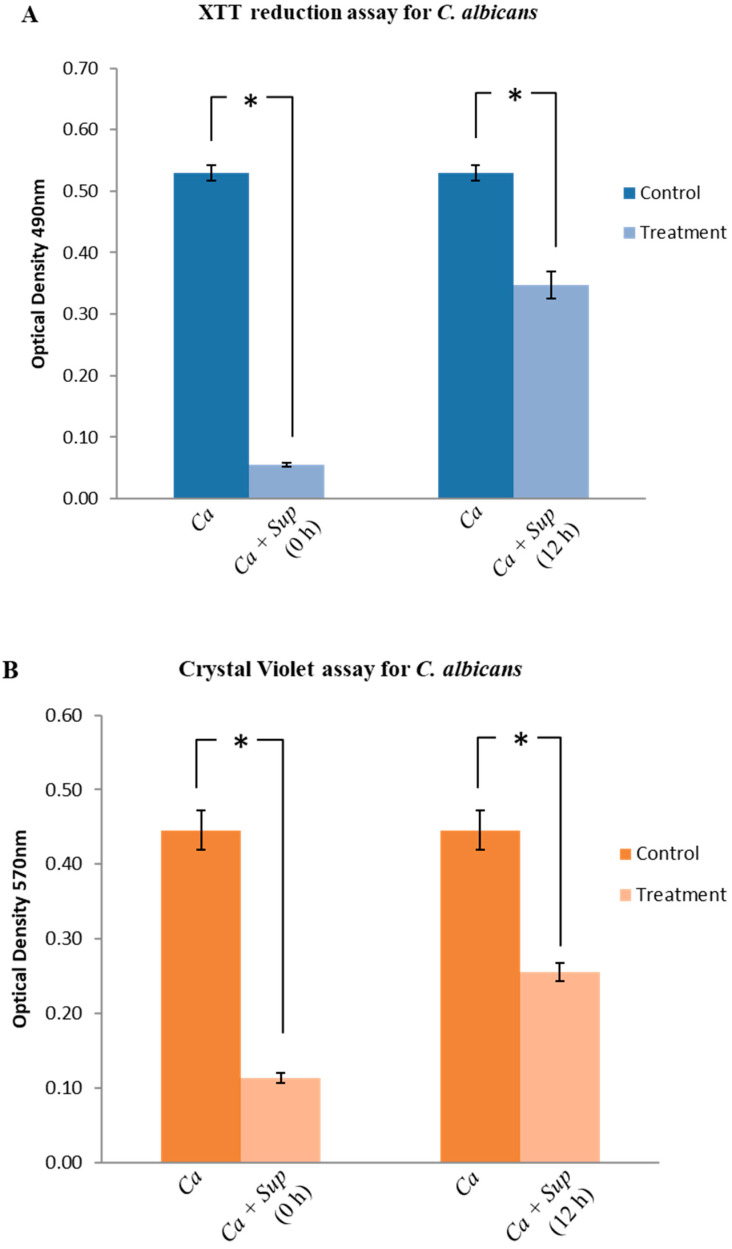
*Lactobacillus plantarum 108* supernatant (Sup) inhibited *C. albicans* (Ca) biofilm formation and demonstrated therapeutic activity against preformed biofilms. (**A**) XTT reduction assay, (**B**) Crystal Violet assay, and (**C**) Colony forming unit counting (CFU) were used for the quantification of biofilms treated with the supernatant at 0 h for the preventive assay and at 12 h for the therapeutic assay. Data are presented as mean + SD and statistical significance (*) was evaluated with respect to the untreated group (*p* < 0.05).

**Figure 4 antibiotics-09-00478-f004:**
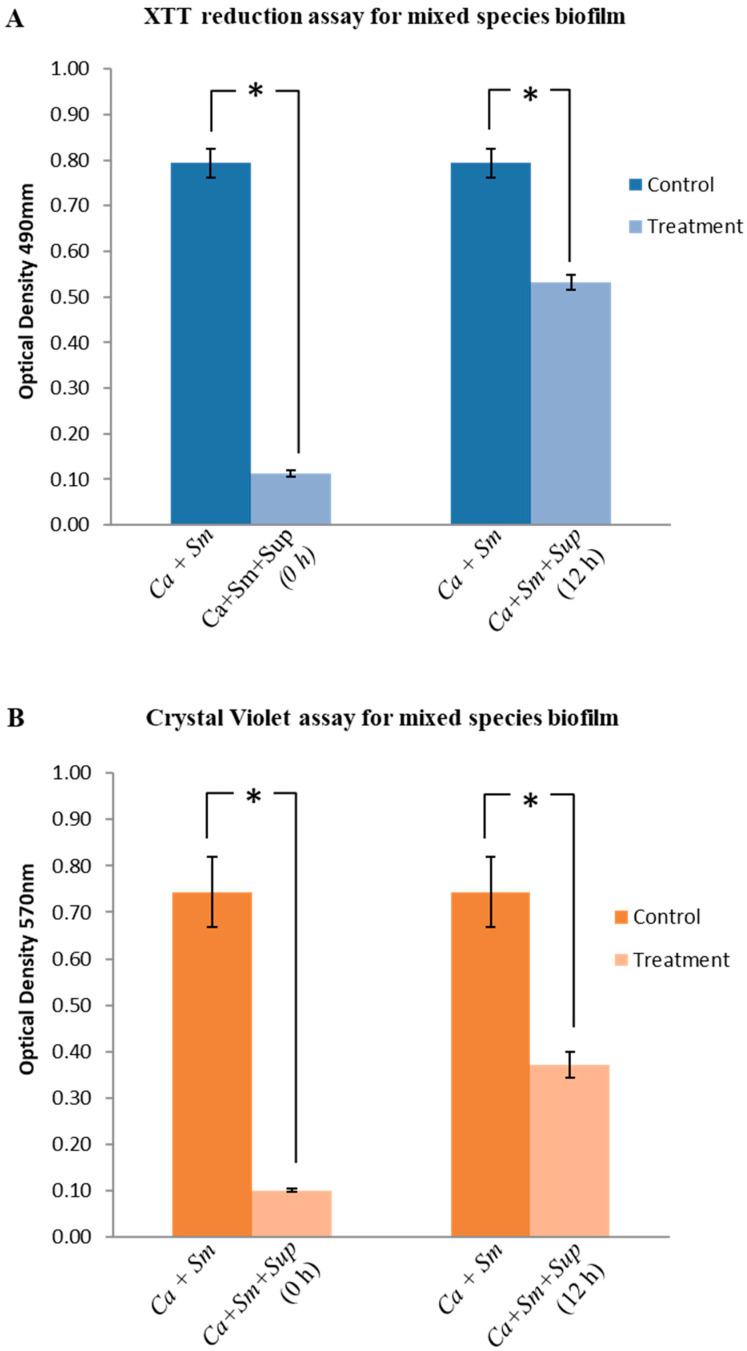
*Lactobacillus plantarum 108* supernatant (Sup) inhibited *S. mutans* (Sm) and *C. albicans* (Ca) mixed species biofilms. The supernatant was introduced into the mixed-species biofilm at 0 h (preventive assay) and at 12 h (therapeutic assay). Biofilms were quantified using (**A**) XTT reduction assay, (**B**) Crystal Violet assay and Colony forming unit counting (CFU) for (**C**) *S. mutans*, and (**D**) *C. albicans.* Data are presented as mean ± SD and statistical significance (*) was evaluated with respect to the untreated group *(p* < 0.05).

**Figure 5 antibiotics-09-00478-f005:**
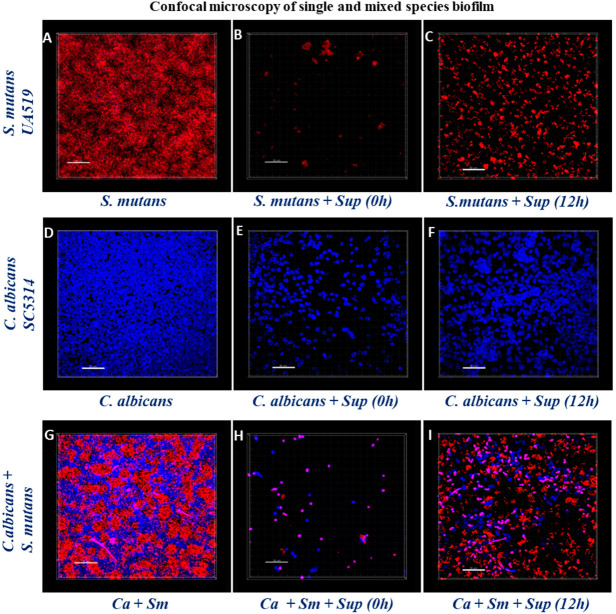
CLSM examination of *S. mutans* (Sm) and *C. albicans* (Ca) single and mixed-species biofilm formation in the presence of *Lactobacillus plantarum 108* supernatant (Sup). The biofilms were fixed and stained with calcofluor white for *C. albicans* (in blue) and propidium iodide for *S. mutans* (in red). Images **A**–**C** shows *S. mutans* single-species biofilms (**A**) control group, (**B**) treated with supernatant at 0 h, (**C**) preformed biofilm treated with the supernatant at 12 h. Images **D**–**F** shows *C. albicans* single-species biofilms (**D**) control group, (**E**) treated with supernatant at 0h, (**F**) preformed biofilm treated with the supernatant at 12 h. Images **G**–**I** shows *S. mutans* and *C. albicans* mixed-species biofilms (**G**) Control group, (**H**) treated with supernatant at 0 h, (**I**) preformed biofilm treated with the supernatant at 12 h.

**Figure 6 antibiotics-09-00478-f006:**
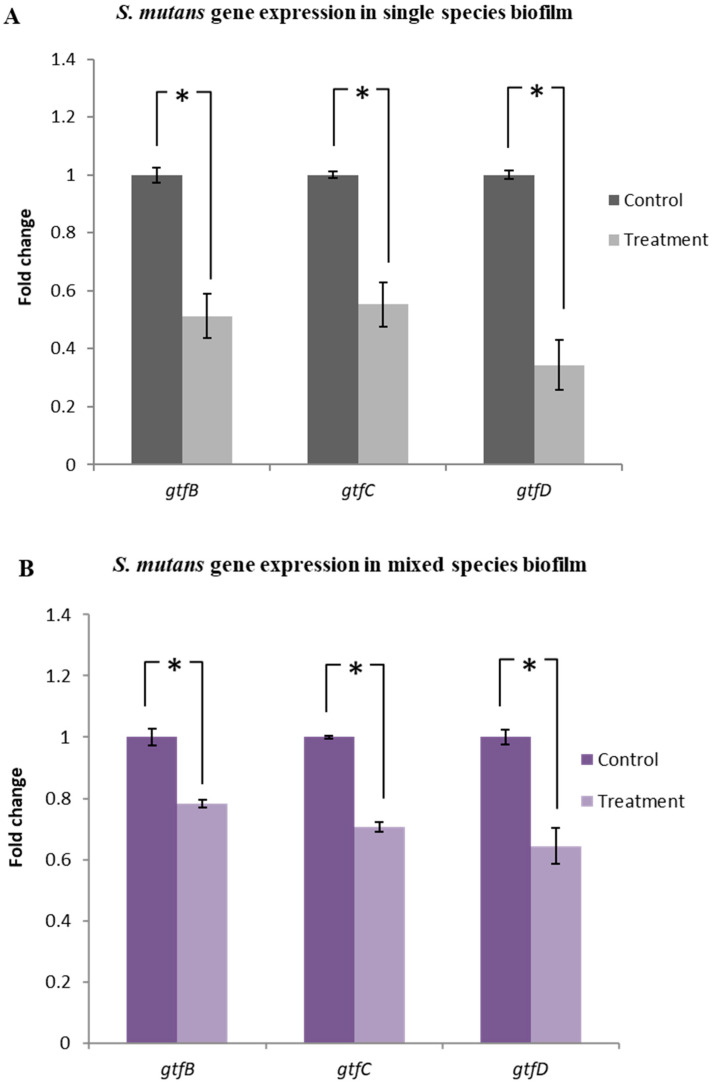
*Lactobacillus plantarum 108* supernatant down-regulated the *gtf* gene expression in *Streptococcus mutans* single and mixed-species biofilms. Quantitative PCR was performed to evaluate the expression of *gtfB, gtfC*, and *gtfD* genes in (**A**) single and (**B**) mixed species biofilms treated with Lp108 supernatant. Target genes in the sample were normalized with the *S. mutans* housekeeping gene 16sRNA. The graphs show the relative expression fold change of treated group with respect to the control group. Data represent the mean ± SD of three independent experiments and statistical significance (*) with respect to the control group (*p* < 0.05).

**Figure 7 antibiotics-09-00478-f007:**
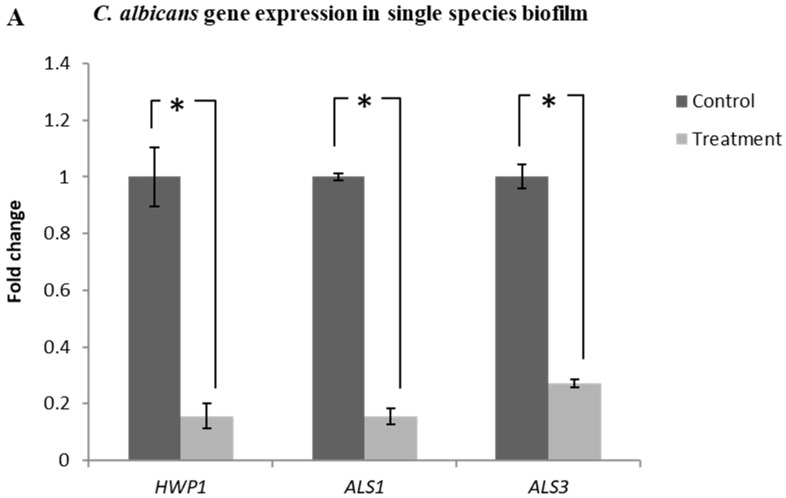
*Lactobacillus plantarum 108* supernatant down-regulated the expression of *HWP1, ALS1* and *ALS3* genes in *Candida albicans* single and mixed-species biofilms. Quantitative PCR helped to evaluate the expression of *HWP1, ALS1* and *ALS3* genes in *C. albicans* (**A**) single and (**B**) mixed species biofilms treated with Lp108 supernatant. Target genes in samples were normalized with the *C. albicans* housekeeping gene *PMA1*.The graphs show the relative expression fold change of treated group with respect to the control group. In the presence of supernatant, expression of hyphal growth associated genes were significantly down-regulated in both single and mixed-species biofilms. Data represent the mean ± SD of three independent experiments and statistical significance (*) with respect to control group (*p* < 0.05).

**Table 1 antibiotics-09-00478-t001:** *Streptococcus mutans* forward (F) and reverse (R) primers used for qRT-PCR [31].

Primer Sequence Used for *Streptococcus mutans*		
Gene	Primer Sequence (5’-3’)	
***16sRNA***	F:	CCT ACG GGA GGC AGC AGT AG
	R:	CAA CAG AGC TTT ACG ATC CGA AA
***gtfB***	F:	AGC AAT GCA GCC AAT CTA CAA AT
	R:	ACG AAC TTT GCC GTT ATT GTC A
***gtfC***	F:	GGT TTA ACG TCA AAA TTA GCT GTA TTA GC
	R:	CTC AAC CAA CCG CCA CTG TT
***gtfD***	F:	ACA GCA GAC AGC AGC CAA GA
	R:	ACT GGG TTT GCT GCG TTT G

**Table 2 antibiotics-09-00478-t002:** *Candida albicans* forward (F) and reverse (R) primers used for qRT-PCR [65].

Primer Sequence Used for *Candida albicans*		
Gene	Primer Sequence (5’-3’)	
***PMA1***	F:	TTGAAGATGACCACCCAATCC
	R:	GAAACCTCTGGAAGCAAATTCG
***HWP1***	F:	GCTCAACTTATTGCTATCGCTTATTACA
	R:	GACCGTCTACCTGTGGGACAGT
***ALS1***	F:	GAC TAG TGA ACC AAC AAA TAC CAG A
	R:	CCA GAA GAA ACA GCA GGT GA
***ALS3***	F:	AATGGTCCTTATGAATCACCATCTACTA
	R:	GAGTTTTCATCCATACTTGATTTCACAT

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
