# Peer review of "Lactobacillus Plantarum 108 Inhibits Streptococcus mutans and Candida albicans Mixed-Species Biofilm Formation"

_antibiotics, 2020, doi:10.3390/antibiotics9080478_

Round 1
Reviewer 1 Report
"Lactobacillus plantarum 108 inhibits Streptococcus mutans and Candida albicans mixed-species biofilm formation”
The current manuscript describes an efficacy of the secretory supernatant of a probiotic strain, Lactobacillus plantarum 108 against S. mutans and Candida mixed-species biofilms. The study focuses on an interesting topic. The concept of using Lactobacillus species for disease treatment and prevention as well as health restoration and maintenance is not new. However, in recent times, there has been a renewal of interest in the use of probiotics, especially as alternative methods of treatment and prevention of bacterial and fungal infections. S. mutans have been reported as an acidogenic microorganism forming a virulent biofilm on the tooth surface in the case of ECC. According to previous studies S. mutans creates biofilm with the human fungal pathogen Candida albicans.
The manuscript is correctly written, novel, and it could be of scientific and practical value for scientific community. The abstract is clear and informative, the results are clearly presented and conclusion is logical and scientifically well documented.
Detailed comments are outlined below:
In opinion of the reviewer, the manuscript contains too little information about ECC (causes, symptoms, role of bacterial biofilm). Moreover, there is not enough data on S. mutans strain and Candida albicans strain used in the study (e.g. the source of origin).
Did the authors carry out a control test using reference indicator microorganisms?
In order to achieve a probiotic effect, it is necessary to use the appropriate number of microorganisms, which is given as the minimum number of colony forming units (CFU). The manuscript does not give the CFU value for Lactobacillus plantarum.
All abbreviations need to be clearly defined at first use. For example, XTT assay, CV assay,…..
Author Response
Comment 1:In opinion of the reviewer, the manuscript contains too little information about ECC (causes, symptoms, role of bacterial biofilm).
Response 1:We apologize for missing out the details on ECC. Information about causes, symptoms and role of bacterial biofilm in ECC have been added in the manuscript in page 1 and 2, lines 43 to 48.
Comment 2:Moreover, there is not enough data on S. mutansstrain and Candida albicans strain used in the study (e.g. the source of origin).
Response 2:For the study, we used S. mutansand C. albicansstrains that are commonlyused inlaboratories working on oral microbiology. The S. mutansUA159, serotype C was originally isolated from a child with active caries in 1982and C. albicansSC5314 strain used in the study is identified as a clinic strain originally isolated from a patient with generalized Candida infection. Information on source of origin for S. mutansand C. albicansare updated in section 4 (Materials and Methods), sub-section 4.1.Please see page 14, lines 295-298.
Comment 3:Did the authors carry out a control test using reference indicator microorganisms?
Response 3:We thank the reviewer for bringing up the point. We agree that it is important to test the activity of the L. plantarum 108supernatant against other reference strains to determine the efficacy. However, in the present study we intend to specifically studythe inhibition of oral bacteria and fungi associated with early childhood caries (ECC). Therefore, we selected S. mutansand C. albicans, which are the most common and extensively studied bacterium and the fungus associated with ECC. In a future study which would characterize the supernatant, we will include more reference microorganisms for testing.
Comment 4:In order to achieve a probiotic effect, it is necessary to use the appropriate number of microorganisms, which is given as the minimum number ofcolony forming units (CFU). The manuscript does not give the CFU value for Lactobacillus plantarum.
Response 4: We agree with the reviewer that it is necessary to use the appropriate number of microorganisms. In this study, we used the cell-free supernatant of Lactobacillus plantaruminstead ofthe Lactobacillus plantarumbacteria itselfto test the inhibitory activity. However, we used a stringent protocol for thepreparation of the L. plantarumsupernatantto obtainsimilar results between batches of supernatants used for the experiment. L. plantarumloopful culture was taken from a fresh agar plate, inoculated in MRS broth and cells were grown for 18 h at 37° C. This overnight culture was subjected to centrifugation (4000×g, 10 min, 4° C) following which, it was washed twice with PBS. The optical density of the standard cell suspension was adjusted to 0.5at a wavelengthof 600 nm using a spectrophotometer (UV-1700 Shimadzu, Tokyo, Japan). For supernatant preparation, this standard cell suspension was diluted 1:100 in MRS broth and further incubated for 24 h at 37° C. Following 24 h of incubation, the bacterial culture wassubjected to centrifugation (4,000xg for 10 min, 4°C) and subsequently filtered through sterilized 0.22 μm pore size membrane (Surfactant-free cellulose acetate, Ministart syringe filter, Sartorious, Singapore) and supernatant was collected. This freshly prepared cell free supernatant was used to check the inhibitory activity.The detailed methodology for supernatant preparation is mentioned in page 15, lines 306-312.
Comment 5:All abbreviations need to be clearly defined at first use. For example, XTT assay, CV assay
Response 5:We would like to thank the reviewer for pointing this out. In the revised version, all the abbreviations have been clearly defined at the
Reviewer 2 Report
This paper reports the activity of the supernatant of the Lactobacillus plantarum 108 culture. The supernatant of the culture showed antibacterial and biofilm formation inihibition activities against Streptococcus mutans and Candida albicans. Similar activities were observed against the mixture of Streptococcus mutans and Candida albicans. The authors also indicated that the supernatant of this strain inhibits the genes related to the biofilm formation. However, the without the identification of the actual compound related to the activities, the information in the manuscript is too general for publication.
Line 102, please add reference for XTT reduction assay, CV assay.
Line 298, please add reference or the detail of the preparation of MRS broth.
Line 301, please add reference or the detail of the preparation of CFTYE broth.
Line 346, please add reference or the detail of the preparation of GMM broth.
Author Response
Comment 1: This paper reports the activity of the supernatant of the Lactobacillus plantarum 108 culture. The supernatant of the culture showed antibacterial and biofilm formation inhibition activities against Streptococcus mutans and Candida albicans. Similar activities were observed against the mixture of Streptococcus mutans and Candida albicans. The authors also indicated that the supernatant of this strain inhibits the genes related to the biofilm formation. However, without the identification of the actual compound related to the activities, the information in the manuscript is too general for publication Response 1: We thank the reviewer for finding interest in our work. Most of the work in this study was focused on finding a potential probiotic bacterium whose secretory components would be of practical and scientific significance in treatment of bacterial and fungal infections. Kindly note that this is the first study that aims at evaluating the efficacy of the Lactobacillus plantarum 108 supernatant, against S. mutans and C. albicans single and mixed-species biofilms. We agree with the reviewer that in the context of this interesting finding, further efforts are needed to fully uncover the molecular mechanism of the supernatant of Lactobacillus plantarum 108 that inhibits S. mutans and C. albicans single and mixed-species biofilms. To gain further insight of L. plantarum supernatant mediated inhibition of S. mutans in single and mixed species biofilms, we analyzed the expression of genes associated with the glucosyltransferase activity (gtfB, gtfC and gtfD) of S. mutans which play a crucial role in the biofilm formation. Similarly, the expression of hyphal growth associated Candida genes HWP1, ALS1 and ALS3 were also examined in the presence of probiotic supernatant. Gene expression results were found to be consistent with the inhibitory activity of probiotic supernatant as shown by biofilm quantification assays and confocal imaging. We believe that these findings will lay down a foundation for future studies to unravel the exact antimicrobial compound/s that this probiotic supernatant contains which inhibits S. mutans and C. albicans biofilm formation. We have included a statement to acknowledge the importance of studying the exact antimicrobial compound/s responsible for the inhibition, which would be performed in a future study. Please see page 14, lines 288-291.
Comment 2: Please add reference for XTT reduction assay, CV assay. Response 2: We would like to apologize for missing out these details. References for XTT and CV reductions are available in sub-sections 4.5.1 and 4.5.2, respectively. Please see page 15 line 342 and page 16 line 349.
Comment 3: Please add reference or the detail of the preparation of MRS broth. Response 3: We have added the details of MRS broth preparation in the revised version of manuscript, in sub-section 4.2 (page 15, lines 306-310). As also pointed out by reviewer 3, details of material source and brands have also been added.
Comment 4: Please add reference or the detail of the preparation of UFTYE broth Response 4: We thank the reviewer for highlighting this shortcoming. In the revised version of the manuscript, references for UFTYE preparation has been added in sub-section 4.3. Please see page 15, line 318.
Comment 5: Please add reference or the detail of the preparation of GMM broth Response 5: In the revised version of manuscript, details on preparation of GMM agar used for CFU counting, has been added in sub-section 4.5.3. Please see page 16 line 356-358.
Reviewer 3 Report
The paper is really well presented, well written and easy to read. I only have a few comments regarding the presentation.
Page 3: It would be better use the same scale in the two graph.
Page 3, line 102: it would be better to write in full XTT, CV and CFU.
pages 4 to 11: it would be better add the "title" on top of the graphs and the % of reduction.
pages 5 and 8: why with CFU test the % of reduction was low?
page 8: it would be better to write in full CLSM.
page 13: materials and methods section lacks of code and brand of the products used.
page 14 line 330: it would be better to write in full XTT.
Author Response
Comment 1: Page 3; It would be better use the same scale in the two graph. Response 1: We thank reviewer for this suggestion. In the revised manuscript, the scales for the graphs for planktonic growth reduction of S. mutans and C. albicans have been updated to the same scale.
Comment 2: Page 3, line 102; it would be better to write in full XTT, CV and CFU. Response 2: Noted on the recommended changes. And as also pointed out by reviewer 1, full form for XTT, CV, CFU and CLSM have been mentioned at the first usage in the revised manuscript. Please see page 3, lines 107-109.
Comment 3: Pages 4 to 11: it would be better add the "title" on top of the graphs and the % of reduction. Response 3: As recommended, titles have been added to all the figures. And to maintain the aesthetics of the figures, the detailed interpretation including the percentage reductions for both control and preventive groups have been mentioned in the text following the respective figures.
Comment 4: Pages 5 and 8: why with CFU test the % of reduction was low? Response 4: We agree with the reviewer that in the CFU test, the percentage reduction was less/different from the XTT assay or the CV assay. Although these three assays are used for biofilm quantification, we understand that the parameters measured in each assay is different. For example: CV assay measures the total biomass including all the cells and the extracellular matrix of the biofilm while XTT assay measures the metabolic activity of cells in the biofilm. CFU measures the number of viable cells in the biofilms. Therefore, it is possible that we see certain changes in the percentages. However, the overall results from the the XTT, CV and CFU counting complemented to each other as well as to the Confocal Laser Scanning Microscopy (CLSM) data. These evaluations showed a noticeable reduction in the treatment group compared to the control. Based on these results, we concluded that L. Plantarum 108 supernatant inhibits both single and mixed species biofilm.
Comment 5: Page 8, it would be better to write in full CLSM. Response 5: Noted on the suggestions. The full form of CLSM has been updated in the revised manuscript in the subsection 2.5. Please see page 10 line 172.
Comment 6: Page 13, materials and methods section lacks of code and brand of the products used. Response 6: We would like to thank the reviewer for highlighting this shortcoming. Few of the missing brands and codes for the materials used have been updated in the revised manuscript in section 4.
Comment 7: Page 14 line 330: it would be better to write in full XTT. Response 7: As also pointed out by reviewer 1, full form of XTT has been updated in the revised manuscript at the first usage. Please see page 3, line 107-10
Round 2
Reviewer 2 Report
As I mentioned before, without the identification of the compounds with the activity, I believe that the study is not significant enough for publication. This part was not improved.
The other part seemed to be improved.
Thus, if the other referees are satisfied with the paper, I will recommend this paper for publication.